# Characterization, Selection, and Trans-Species Polymorphism in the MHC Class II of Heermann’s Gull (Charadriiformes)

**DOI:** 10.3390/genes13050917

**Published:** 2022-05-20

**Authors:** Misael Daniel Mancilla-Morales, Enriqueta Velarde, Araceli Contreras-Rodríguez, Zulema Gómez-Lunar, Jesús A. Rosas-Rodríguez, Joseph Heras, José G. Soñanez-Organis, Enrico A. Ruiz

**Affiliations:** 1Departamento de Zoología, Escuela Nacional de Ciencias Biológicas, Instituto Politécnico Nacional, Prolongación de Carpio y Plan de Ayala s/n, Col. Santo Tomás, Ciudad de Mexico CP 11340, Mexico; 2Instituto de Ciencias Marinas y Pesquerías, Universidad Veracruzana, Hidalgo 617, Colonia Río Jamapa, Boca del Rio, Veracruz CP 94290, Mexico; enriqueta_velarde@yahoo.com.mx; 3Departamento de Microbiología, Escuela Nacional de Ciencias Biológicas, Instituto Politécnico Nacional, Prolongación de Carpio y Plan de Ayala s/n, Col. Santo Tomás, Ciudad de Mexico CP 11340, Mexico; aracelicontreras21@gmail.com (A.C.-R.); zgomezl@ipn.mx (Z.G.-L.); 4Departamento de Ciencias Químico-Biológicas y Agropecuarias, Universidad de Sonora, Lázaro Cárdenas del Río No. 100, Francisco Villa, Navojoa CP 85880, Mexico; jesus.rosas@unison.mx; 5Departament of Biology, California State University, San Bernardino, 5500 University Parkway, San Bernardino, CA 92407, USA; joseph.heras@csusb.edu

**Keywords:** duplication, Gulf of California, *Larus heermanni*, peptide-binding sites, seabirds

## Abstract

The major histocompatibility complex (MHC) enables vertebrates to cope with pathogens and maintain healthy populations, thus making it a unique set of loci for addressing ecology and evolutionary biology questions. The aim of our study was to examine the variability of Heermann’s Gull MHC class II (MHCIIB) and compare these loci with other Charadriiformes. Fifty-nine MHCIIB haplotypes were recovered from sixty-eight Heermann’s Gulls by cloning, of them, twelve were identified as putative true alleles, forty-five as unique alleles, and two as pseudogenes. Intra and interspecific relationships indicated at least two loci in Heermann’s Gull MHCIIB and trans-species polymorphism among Charadriiformes (coinciding with the documented evidence of two ancient avian MHCIIB lineages, except in the Charadriidae family). Additionally, sites under diversifying selection revealed a better match with peptide-binding sites inferred in birds than those described in humans. Despite the negative anthropogenic activity reported on Isla Rasa, Heermann’s Gull showed MHCIIB variability consistent with population expansion, possibly due to a sudden growth following conservation efforts. Duplication must play an essential role in shaping Charadriiformes MHCIIB variability, buffering selective pressures through balancing selection. These findings suggest that MHC copy number and protected islands can contribute to seabird conservation.

## 1. Introduction

The major histocompatibility complex (MHC) is a multi-gene family that encodes for the essential transmembrane glycoproteins involved in antigen recognition and presentation as part of the adaptive immune response within vertebrates [1,2]. Duplication [3], gene conversion [4], mutations [5], genetic drift [6], sexual selection [7], environmental variation [8], and demographic history [9] are evolutionary forces and processes that have been proposed to explain MHC polymorphism within populations. Although all of these mechanisms and processes have been extensively studied, their relative importance is still debated, along with the significance of variation across taxa or within populations of the same taxon [10,11].

There are two common class types of MHC protein molecules: class I (MHCI) and class II (MHCII) [12,13]. MHCI spans the membrane of all nucleated cells and responds to intracellular pathogens such as viruses, while MHCII is expressed in antigen-presenting cells and is involved in the adaptive immune response against extracellular pathogens such as bacteria [12,14]. Both classes display a peptide-binding region (PBR), responsible for the presenting of peptides to TCD8+ and TCD4+ lymphocytes [13,15]. Due to the high specificity of this region, it is hypothesized that individuals with more alleles must be able to recognize and deal with more pathogens [16,17]; thus, the higher the polymorphism the better the response against selection pressures [18,19]. Consequently, high polymorphism maintenance in MHC genes is the result of balancing selection mediated by pathogen resistance [20], frequency-dependent selection [21], heterozygote advantage [22], fluctuating selection [23], or some combination of all of them [24]. Nevertheless, the highest pathogen resistance could be attained by an optimal intermediate number of variants rather than the maximum possible [3,11,25].

The first MHC studies in vertebrates were performed on red junglefowl (*Gallus gallus*), commonly known as the domestic chicken, and although recent advancements are emerging for non-model birds, there are still many more studies in non-birds [3,26,27]. Chickens have a smaller MHC (in both sequence size and gene copy number) than humans and a rarer occurrence of pseudogenes; thereby, the minimal essential MHC hypothesis has been proposed for birds [28,29]. It suggests that few homologues (or orthologues) of MHC genes between birds and mammals have been conserved over evolutionary times [28]. Conversely, the genome-wide duplication hypothesis (the appearance of many duplicate genes without essential functions attributed to genome-wide duplication events occurring at the base of vertebrates) is widely accepted to explain the evolution of the adaptive immune system [30,31]. The avian MHC possesses substantial diversity in structure, gene number, and intron length among species, and varies from a single gene to extremely high gene duplication with tens of loci in some passerines [30,31,32,33]. It is still unknown whether a few copies in the compact MHC organization are ancestral traits in birds, and how their variability has evolved into its current complexity [31,32]. Nevertheless, exon 2 of the MHC class II shows a characteristic polymorphism with hypervariable segments occurring in different combinations over different alleles, probably resulting from duplications with neofunctionalization [12,30,31,32,33].

In studies of the evolution of MHC, concerted evolution is considered responsible for the diversity observed, even though it can complicate phylogenetic reconstructions by hidden orthologous relationships [5,30]. Polymorphisms maintained within closely related species over species radiation supports that MHCII genetic variation and balancing selection may persist throughout speciation processes [10,33]. Thus, some MHCII alleles may be more similar between species than within species [11,34]. Within birds (Aves), this could be explained by a duplication event prior to the evolution of all the extant birds (over 100 million years ago), which may have been masked by concerted evolution [30]. Available evidence indicates that a single common evolutionary pattern is very unlikely in all birds, given the contrasting differences in genetic diversity displayed in all species studied [30,31,32,33,34]. However, a global analysis at the avian MHC elucidated a stronger selection in MHC class II genes in non-passerines than in passerines [35]. Thus, limited MHCII polymorphism may raise conservation concerns as their variability is associated with fitness and adaptation [36].

Despite the recent advances in genomics, estimates of MHC variability in wild populations are scarce in many avian groups [37,38]. An example of this is the Charadriiformes avian group, as *MHCII* genes in the families Scolopacidae, Alcidae, Charadriidae, and Laridae have been studied and characterized only in a few species [39,40,41,42,43]. Therefore, research on understudied and threatened non-model birds is essential to address questions in the field of evolution and ecology.

This study explores the MHCII variation in Heermann’s Gull (*Larus heermanni*), a threatened seabird that nests mainly on Isla Rasa (Gulf of California, Baja California, Mexico); this island hosts >90% of the world population during the breeding season [44]. Recombination events shape MHC variability in birds [30,31,32,33,34,35,39], and there is evidence of duplication in *MHCII* genes from other species of Charadriiformes [32,33,40,41,42]. Thus, this study also sought to contribute to our current knowledge of polymorphism, functional peptide sites, and *MHCII* gene evolutionary patterns in Charadriiformes. In our analyses, intraspecific relationships in Heermann’s Gull revealed at least two putative MHCII loci, and the most common alleles were modeled by homology based on human and chicken templates. Next, codon-specific signatures of selection were tested and compared among homologous bird and mammal sequences [33,45,46,47]. Next, the highest polymorphic sites were used to investigate the signatures of historical selection in Charadriiformes. Then, neutrality tests and mismatch distribution were used to infer whether the MHCII variability of Heermann’s Gull was consistent with the reported demographic changes [48,49]. Finally, the extent of trans-species polymorphism was addressed by the phylogenetic relationships based on the currently known MHCII sequences of Charadriiformes seabirds.

## 2. Materials and Methods

### 2.1. Sampling, DNA Extraction, and PCR Amplification of MHCIIB

Heermann’s Gull individuals included in this study were sampled on Isla Rasa (28°49′28″ N, 112°58′50″ W) and Isla Cardonosa (28°53′16″ N, 113°01′51″ W) in 2011 and 2012 using walk-in traps (specific details on sample collection set up are explained in [49,50]). The blood samples were deposited in the Laboratory of Ecology at ENCB-IPN (Mexico). In this study, we used ~5–10 µL of blood from each gull (*n* = 68 gulls; 66 from different locations on Isla Rasa, and 2 from Isla Cardonosa). Then, total genomic DNA extraction was performed using the DNeasy Blood & Tissue Kit, following the manufacturer’s guidelines (Qiagen, Hilden, Germany). Next, 198 bp of exon 2 of the MHC class II chain B or β1 domain (*MHCIIB* gene) was amplified, including much of the most polymorphic and putatively functional peptide-binding region [51]. For this, pen-1 and pen-4 primers were used, which were designed with well conserved, orthologous *MHCII* genes [52]. Next, both primers were modified to improve their affinity to Heermann’s Gull and named as ‘gull-1’ (5′ AA*T* GG*T* ACC GAG CGG GTG AGG T 3′) and ‘gull-4’ (5′ CCC GTA GTT GT*A*
*CC*G GCA- 3′). PCR reactions were performed in final 10 µL reaction volumes, containing 5 µL of 2× MasterMix (Promega, Madison, WI, USA), 1 µL of each primer (100 mM), and 3 µL of genomic DNA (~25 ng). Thermal conditions were set to reduce the PCR artifacts [53] and consisted of initial denaturation (94 °C, 3 min), followed by a touchdown PCR where the annealing temperature increased 1 °C from 56 to 60 °C (15 s at 94 °C, 30 s at 56–60 °C, and 45 s at 72 °C). Then, 20 additional cycles were run at a constant annealing temperature of 60 °C, followed by a final extension of five minutes at 72 °C. The amplification of the expected size (198 bp) of sequences was confirmed by visualizing 4 µL (~50 ng/µL) of product reaction by electrophoresis on 2% agarose gels.

### 2.2. Cloning, Sequencing, and the Validation of Alleles

PCR products were cloned using the pGEM^®^-T Easy vector (Promega) for each gull, following the manufacturer’s instructions and corroborating the expected insert size. Independent PCR reactions and eight to twenty-four clones were used to obtain all possible haplotypes per individual. Each clone was Sanger-sequenced by capillary electrophoresis with an ABI 3730xl system DNA analyzer using the universal primer T7 (5′ AAT ACG ACT CAC TAT AG 3′). The resulting sequences were edited, aligned, and translated into amino acids through Seaview v5.0.4 [54]. Then, the identity of each sequence was verified using the standard nucleotide search in BLAST via the internet [55]. Haplotypes were resolved in DnaSP v6.12 [56]; alleles found in two or more individuals were considered as putative true alleles, while those found in at least three identical clones but only in an individual were taken as potential unique alleles [57,58]. All alleles were verified with at least two independent PCR runs per individual (Appendix A). Haplotypes containing stop codons were considered pseudogenes. Lastly, all the MHCIIB haplotypes were submitted to the GenBank (MW848537-MW848595).

### 2.3. Intraspecific Relationships and the Characterization of MHCIIB Polymorphisms

To confirm whether the MHCIIB alleles of Heermann’s Gull displayed convergence or reciprocal monophyly, a tree was built via the maximum likelihood (ML) method using 1000 bootstraps and the GTR + G + I substitution model in MEGA X [59]. Next, all the MHCIIB alleles were connected in a minimum spanning network to visualize their frequencies and relationships in POPart 1.7 [60]. Moreover, a phylogenetic network was made using a neighbor-net method and HKY85 genetic distances in SplitsTree v5.2.26-beta [61]. The networks represent evolutionary history as a phylogenetic tree with additional edges where internal nodes represent ancestral traits and nodes with more than two origins correspond to reticular events such as duplication and recombination [61,62].

The *MHCIIB* genetic diversity was assessed for codons located in PBR and non-PBR, as inferred in humans by Brown et al. [47]. Thus, the corresponding number of segregating sites (*S*), the average number of nucleotide differences (*K*), and average nucleotide diversity (π) were calculated using DnaSP v6 [56]. Next, an analysis of selection was performed by estimating the overall mean evolutionary distances for the nucleotide (*d*_nt_) and amino acid (*d*_aa_) sequences using the *p*-distance model and the uniform rates among the sites with standard errors were calculated by 1000 bootstraps in MEGA X [59].

Although several MHC studies are based on peptide-binding sites (PBSs) identified in humans to assess selection, these PBSs may not always be appropriate, as selective forces may act on non-PBSs [35,63]. Therefore, the Wu–Kabat variability coefficient was used to identify the more polymorphic amino acids while, simultaneously, the conserved fragments were explored using the PVS server [64]. In addition, the SWISS-MODEL server was used to compare and visualize the structural differences between putative loci [65]. Finally, the most common alleles were modeled by homology based on the MHC templates of chicken (Protein Database Bank ID: 6 kvm) and human (Protein Database Bank ID: 4h25), both available in the RCSB Protein Data Bank [66].

### 2.4. Analysis of Selection Signatures in Heermann’s Gull and Other Charadriiformes

MHCIIB sequences usually have an excess of non-synonymous (*d*N) over synonymous (*d*S) substitution rates (expressed as ω = *d*N/*d*S) at PBR codons due to the selection imposed by pathogens which is, namely, positive (diversifying) selection (ω > 1). In contrast, synonymous nucleotide substitutions *d*S are expected to be higher than non-synonymous substitutions *d*N at non-PBR codons, which may be part of the functional and conserved structures under negative (purifying) selection (ω < 1). Nonetheless, under neutrality, similar substitution rates are expected in both PBR and non-PBR codons (ω = 1) [6,20,36,63,67]. To infer whether these sites were under either diversifying or purifying selection in Heermann’s Gull and other Charadriiformes available in GenBank, we used Bayesian and maximum likelihood methods implemented by the HyPhy package in the DataMonkey server [68,69].

First, the conservative fixed effects likelihood (FEL) method was used to estimate the synonymous (α) and non-synonymous (β) substitution rates site-by-site through 66 codons for the 2 putative MHCIIB loci inferred in Heermann’s Gull (DAB = 23 alleles and DBB = 34 alleles) (see Appendix A). This method is suitable for low to intermediate datasets, and sites with thresholds of *p* < 0.1. It has reliable estimates of selection if the null hypothesis (α = β) is rejected [70]. Then, FEL was employed to search sites under positive or negative selection across the whole phylogeny per species, assessing each site’s likelihood using 100 bootstraps. Second, codon-specific signatures of positive and negative selection were determined for Laridae and Charadriiformes using fast unconstrained Bayesian approximation (FUBAR) [71]. FUBAR is suited for larger datasets, assumes a constant selection pressure for each site along the whole phylogeny, is less restrictive, and has more power than FEL [70,71]. FUBAR employs a Markov Monte Carlo Chain (MCMC) and the sites under selection are well supported at a threshold of >0.9 Bayesian posterior probabilities [71]. Third, to infer sites under episodic diversifying selection, we employed the mixed effects model of evolution (MEME) with a *p*-value threshold of 0.1 [72]. MEME uses maximum likelihood assuming a background *d*N/*d*S to determine whether *d*N is greater than the background rate site-by-site, and false positive rates are well controlled even in strict neutrality [72].

The FUBAR and MEME approaches were used for the family Laridae and the order Charadriiformes, separately. For the bird family Laridae, we analyzed 237 sequences of Heermann’s Gull, Black-legged Kittiwake (*Rissa tridactyla*), the Ivory Gull (*Pagophila eburnea*), and the Common Tern (*Sterna hirundo*) (see Appendix A). For the bird order Charadriiformes, 392 sequences of fourteen species were analyzed including the families: Alcidae (Least Auklet *Aethia pusilla*, Crested Auklet *Aethia cristatella*, Marbled Murrelet *Brachyramphus marmoratus*, Razorbill *Alca torda*, Atlantic Puffin *Fratercula arctica*, and Common Murre *Uria aalge*), Laridae (the four previously mentioned species), Scolopacidae (Great Snipe *Gallinago media*, Black-tailed Godwit *Limosa limosa*, and Ruff *Philomachus pugnax*) and Charadriidae (Snowy Plover *Charadrius nivosus*) (see Appendix A). All the analyzed MHCIIB sequences from other studies were downloaded and are available in GenBank.

To contextualize the genetic variation and historical selection found in Heermann’s Gull, MHCIIB haplotypes were compared with homologous ones from other Charadriiformes (all the species mentioned above, see Appendix A). The analyses were carried out for all codons, PBSs, and non-PBSs, considering twenty sites under diversifying selection inferred in this study by both FUBAR and MEME approaches (see the results section). To compare the genetic variation, synonymous (*dS*) and non-synonymous (*dN*) substitution rates were calculated through the modified Nei–Gojorobi method with Jukes–Cantor correction and the G + I model using 1000 bootstraps in MEGA X [59]. Similarly, historical selection was tested for adaptive changes indicated by neutral (*dN* = *dS*), deleterious (*dN* < *dS*), and advantageous (*dN* > *dS*) non-synonymous mutations through z-tests performed in MEGA X [59].

In addition, the RELAX method was used to test for the relaxation or intensification of selection pressures on the MHCIIB through comparative phylogenetic trees [73]. The alleles of Heermann’s Gulls from this study (Appendix A) were considered as the test dataset and compared separately with the alleles from each of the thirteen Charadriiformes species as the reference dataset (Appendix A). RELAX allows the identification of trends in the stringency of natural selection: a significant result of k > 1 indicates intensified selection while a significant result of k < 1 indicates relaxed selection along the testing branches compared with the background branches [73].

### 2.5. Historical Demographic Changes in Heermann’s Gulls

Nuclear (nuDNA) and mitochondrial (mtDNA) DNA sequences of Heermann’s Gull were compared to infer differences in nucleotide diversity, departures from neutrality and demographic changes. For this purpose, we analyzed the MHCIIB variants (nuDNA) recovered in this study and compared them to the cytochrome *b* (mtDNA) sequences already available [49]. To reduce bias in the level of variation affecting the comparisons between nuDNA and mtDNA, we employed isolates from the same individuals in both datasets, replacing 2 individuals to complete 68 gulls’ mtDNA (Appendix A).

To infer standard diversity indices and departures from neutrality we employed the Tajima’s *D* [74], Fu’s *F*_s_ [75], and Ramos-Rozas and Onsins’ *R*_2_ [76] tests by 10,000 simulations in DnaSP v6 [63]. Tajima’s *D* is based on estimates of the number of segregating sites and the mean pairwise differences between sequences, while Fu’s *F*_s_ considers the number of different haplotypes in the sample [77]. In contrast, *R*_2_ compares the differences among singleton mutations and the average number of nucleotide differences, with low values expected under a recent large population growth [76,77]. Significant negative values for Tajima’s and Fu’s statistics indicate the low frequency of mutations; therefore, they are consistent with demographic expansion and/or purifying selection [74,75]. At the same time, non-significant positive values for Tajima’s *D*, Fu’s *F*_s_, and Ramos-Rozas and Onsins’ *R*_2_ indicate neutral selection or a mutation-drift equilibrium [74,75,76].

A mismatch distribution analysis was carried out through a generalized least-square approach using 10,000 bootstraps in Arlequin version 3.5.2 [78] to distinguish between a demographically stable population and one that has had recent demographic expansion. Usually, the distribution of differences among pairs of haplotypes is multimodal in populations at the demographic equilibrium, while unimodal distributions are typical in populations undergoing demographic expansion [79]. To test the goodness of fit for demographic expansion, we employed the sum of square deviations (SSD) between the observed and expected frequencies of pairwise comparisons [80]. Additionally, the raggedness index (*rg*) of the observed distribution was used. *rg* shows larger values for multimodal distributions in a stationary population than for unimodal distributions observed in expanding populations [76,81]. Mismatch distributions were used to estimate the demographic parameters tau (τ = 2*ut*) for cytochrome *b* and MHCIIB separately (where *u* = cumulative substitution rate and *t* = number of generations since the expansion) from a population size *θ*_0_ (before the expansion) to *θ*_1_ (after the expansion), as well as their confidence intervals by randomization of the data. The values of time-since-expansion (*t*) were estimated from the parameter tau (*t* = τ/2*u*) using the substitution rates of 1.8% Ma^−1^ and 2.0% for cytochrome *b* [49]. In contrast, values of *t* for MHCIIB were estimated using the average adjusted rates of zebra finch (*Taeniopygia guttata*) (0.00160 per site/million years) and chicken (0.00177 per site/million years) [82] (see Appendix A).

### 2.6. Interspecific Phylogenetic Relationships of MHCIIB in Larids and Other Charadriiformes

A common way to identify TSP that avoids confusion with evolutionary convergence is to compare clustering patterns of codons under positive selection versus neutral codons retaining conserved signatures of ancestry [63,83]. Therefore, to distinguish between convergence or TSP among Charadriiformes species, phylogenetic trees were constructed separately to visualize the relationships among codons under diversifying selection (and their counterpart) inferred in this study. Thus, phylogenetic relationships were assessed using the automatic selection of substitution model [84], subtree pruning and regrafting (SPR) tree searching [85], and aLRT SH-like branch support [86], all implemented in PhyML 3.0 [87]. Alleles similar or identical by descent should retain an ancestry signature at neutral sites, while convergence is inferred if alleles between species are more similar at positively selected sites than at neutral sites [5,24,45,88]. To mitigate the bias produced by sample size and sequence length, we analyzed subsets of data restricted to no more than ten different alleles per species (when large available datasets were compared) and discarded alleles without sufficient length (<198 bp after trimming to fit Heermann’s Gull MHCIIB alleles). The B-LB2 MHC from the chicken sample was employed as an outgroup (GenBank Accession: AY744349).

Finally, the phylogenetic relationships for the MHCIIB alleles found in Heermann’s Gull were compared for all codons (antigen-binding and non-antigen-binding sites) within homologous sequences from Charadriiformes. Moreover, we separately drew Laridae’s relationships to obtain a more detailed identification of duplication events through the allelic lineages. For this purpose, we constructed phylogenetic networks using a neighbor-net method based on HKY85 genetic distances implemented in SplitsTree v5.2.26beta [61,62].

## 3. Results

### 3.1. MHCIIB Recovered by Cloning in Heermann’s Gull

We were able to clone 198 bp from MHCIIB exon 2 fragments for all of the 68 studied Heermann’s Gulls. This was the expected number using the modified primers (see Methods; the fragment co-amplification with any other random size was observed in only three individuals and discarded from the analyses). When the sequences of all 198 bp nucleotides were contrasted in BLAST, all showed a high percentage of identity (frequently above 90–95%) with MHCIIB sequences of other Charadriiformes (mainly with the Common Tern and Black-legged Kittiwake). One to six different haplotypes were found per individual, suggesting an intraspecific variation in the number of copies (16 Heermann’s Gulls showed one haplotype, 16 showed two haplotypes, 20 showed three haplotypes, 11 showed four haplotypes, 3 showed five haplotypes, and 2 showed six haplotypes). We successfully recovered 59 MHCIIB haplotypes from 68 Heermann’s Gulls through conventional cloning. Trying to avoid possible PCR artifacts, 12 of these haplotypes were considered true putative alleles (found in two or more individuals), 47 were considered potential unique alleles (found only once in one individual and in at least three independent PCR clones), and 2 haplotypes that encoded for stop codons were considered pseudogenes or non-classical loci. These pseudogenes were used to evaluate the intraspecific relationships (see Appendix A), but were eliminated from the subsequent analyses.

### 3.2. Inferred Duplication by Phylogenetic Relationships within Heermann’s Gull MHCIIB

A maximum-likelihood tree was obtained using twelve putative true MHCIIB alleles of Heermann’s Gull. Intraspecific relationships showed two well-supported clusters (groups of more than two sequences with nodal support of >95) [89], suggesting the presence of at least two loci (Figure 1a). Similarly, when all the fifty-nine haplotypes were analyzed in a separated tree, their relationships showed two groups of sequences, although with lower support (Appendix A). However, low support is not unusual for MHCIIB intraspecific phylogenies and could be typical in recently diverged groups [14,88,89,90,91]. Five putative true alleles and eighteen potential unique alleles could be seen in the cluster with more sequences (DAB, Appendix A). On the other hand, seven putative true alleles and twenty-seven potential unique alleles corresponded to the lower cluster with fewer sequences (DBB, with two pseudogenes also aligning in this group of sequences, Appendix A). Each cluster was dominated by a single common haplotype in cloning: the Lahe-DAB01 allele (14.86%) or the Lahe-DBB01 allele (68.99%), with other alleles ranging in frequencies of less than 1.46%.

In addition, the minimum spanning network showed that the frequency and number of alleles per individual were higher in the DBB cluster than in the DAB cluster. Few mutational changes within each cluster (less than seven steps) suggest recently diverged characters with higher mutational changes among clusters (Figure 1b and Appendix A). The increased presence of a couple of haplotypes with a few others deriving from them may suggest a founder effect, which is typical in populations that have experienced a recent bottleneck [9]. However, another possibility is that the two most common haplotypes were favored by frequency-dependent selection [20,21,22,23,24]. Moreover, the phylogenetic network showed two distant allele groups (Figure 1c and Appendix A) which, together with pseudogenes in the DBB cluster (DBB35 and DBB36 haplotypes in Appendix A), consistently suggests a duplication event. There are studies reporting from two to four loci in Charadriiformes and other avian orders [40,90]. Therefore, both clusters were considered as two different putative loci. Overall, the intraspecific relationships among Heermann’s Gull MHCIIB alleles suggest that the DBB cluster may be more favored by natural selection and might have diverged more recently in time than the DAB cluster.

### 3.3. Characterization Heermann’s Gull MHCIIB Polymorphism

Heermann’s Gull MHCIIB haplotypes were grouped in two clusters regarded as loci. Thus, the genetic variability was examined in all the alleles and for each cluster including putative true and unique alleles but removing haplotypes with stop codons (Table 1). Thus, we observed 55 segregating sites (*S*), 10.4 average nucleotide differences (*K*), and 0.05 average nucleotide diversity (π) in the complete sequences. The DAB cluster showed a lower number of segregating sites (*S* = 28) than the DBB cluster (*S* = 39); however, the DAB cluster showed higher average nucleotide differences and diversity (*K* = 4.98; π = 0.025) than the DBB cluster (*K* = 3.242; π = 0.017). In contrast, the average evolutionary distances of nucleotide (*d*_nt_) and amino acid (*d*_aa_) sequences were higher in the DAB cluster. The PBRs generally showed lower segregating sites than non-PBRs, although they displayed higher average nucleotide differences and evolutionary distances than non-PBRs (Table 1). Overall, the PBRs displayed higher diversity than non-PBRs, suggesting the balancing selection of the PBR residues, but with a stronger effect in the DAB cluster than in the DBB cluster.

The Wu–Kabat variability coefficient (W) showed an average score of 1.99 for all Heermann’s Gull MHCIIB alleles and identified seven highly polymorphic sites: 12, 15, 20, 29, 43, 52, and 53 (Figure 2). All of them corresponded to deduced pockets or depressions within the binding groove (PBR sites) also seen in humans and chickens [46,47] (Figure 3). The DBB cluster (W = 1.53) revealed a slightly higher average variability score than the DAB cluster (W = 1.43), and distinctive variation patterns were found in each cluster. The DAB cluster showed the most amino acid variation at sites 12, and 20, which displayed more than twice the average Wu–Kabat score. For the other nineteen sites, the W values were higher than one. The DBB cluster displayed high polymorphism at sites 12, 15, 29, and 43, with the other 24 sites having W values higher than 1. Patterns of elevated polymorphism correspond with the amino acid identities of chicken BLB chains (β-chain) [45]. Likewise, five fragments of six or more consecutive residues (Figure 2) resembled conserved residues from the chicken BLB main chain interacting with its BLA chain (α-chain) [45]. Overall, the polymorphic sites found corresponded to the PBR sites (see Figure 4) and the largest number of them were found in the DBB rather than in the DAB cluster.

The MHCIIB structure of Heermann’s Gull was modelled for the most common alleles at each putative loci: Lahe-DAB*01 and Lahe-DBB*01 (Figure 3) (results of the assessment are summarized in Appendix A). Using the chicken template 6 kvm from the Protein Data Bank, both alleles showed a higher percentage of identity (Lahe-DAB*01/60.61% and Lahe-DBB*01/59.09%) than the human template 4h25 (Lahe-DAB*01/59.09% and Lahe-DBB*01/56.06%). Conversely, the QMEAN Z-score analysis showed lower values for human (−0.48 to −1.32) than for chicken (−0.92 to −1.44) templates (scores around zero reflect a “native-like” structure); however, this score was deprecated and the GMQE and QMEANDisCo scores were advised for global model quality estimates instead. Thus, for the DAB cluster, the Global Model Quality Estimate (GMQE) showed a slightly higher score for the human than for the chicken template, but for the DBB cluster this was the opposite (Appendix A). Nonetheless, from a large dataset of similar models available in both the SWISS-MODEL [65] and Protein Data Bank [66], the average per-residue QMEANDisCo global score suggests that the chicken template could be more reliable than the human template in both alleles (0.81 ± 0.11 and 0.78 ± 0.11 for Lahe-DAB*01, and 0.80 ± 0.11 and 0.79 ± 0.11 for Lahe-DBB*01, for chicken and human, respectively). Overall, a structural evaluation revealed that both alleles were better modelled by the crystal structure of chicken MHC class II than that of human HLA due to lower MolProbity, higher Ramachandran values, and fewer bad angles.

### 3.4. Analysis of Selection

The 66 codons analyzed in Heermann’s Gull MHCIIB via the maximal likelihood FEL approach revealed a non-synonymous versus synonymous rate ratio higher for the DBB cluster (ω = 1.88; AIC_C_ = 1179.85) than for the DAB cluster (ω = 1.58; AIC_C_ = 1036.47). Both clusters showed several invariant codons and a few sites under purifying and neutral selection; however, no codons with significant diversifying selection were observed (Figure 4a,b). The DAB cluster had only one site with significant purifying selection (codon 46) and neutral selection at twenty-two codons (gray vertical bars, Figure 4a), while the DBB cluster possessed only two sites under significant purifying selection (codons 24 and 47) and twenty-nine sites under neutral selection (gray vertical bars, Figure 4b). Nonetheless, when all the alleles were assessed together by FEL, twenty-one codons displayed neutral selection, three codons displayed diversifying selection, and two codons displayed purifying selection (gray, pink and green vertical bars, respectively, Figure 4c). In contrast, the Bayesian FUBAR approach revealed signals of diversifying selection at five sites (12, 19, 20, 26, 42, and 43) and consistent purifying selection at one site (codon 46) (Figure 4c).

Although the evolutionary history of each species is different, natural groups may also share a pathogen-mediated selection [8,32,34,39,42]. Therefore, we applied different phylogenetic approaches to analyze significant signatures of selection for each species and other groups (Laridae and Charadriiformes). The amino acids under positive selection inferred for the MHCIIB alleles are summarized in Figure 5 and were compared with: (1) amino acids inferred for Laridae and Charadriiformes in this study; (2) the residues for non-passerine birds labeled as putative PBSs by Minias et al. [35]; (3) the antigen-binding groove for humans inferred by Brown et al. [46,47]; and (4) the conserved amino acids inferred from the alignment between the BLB sequences in chickens and representative mammals by Zhang et al. [45].

The application of the maximal likelihood FEL approach at each species revealed significant signatures of selection in a few codons: eight codons were consistently observed in at least four out of fourteen species tested (codons under purifying selection: 1, 3, 24, 36, and 47; codons under diversifying selection: 8, 39, and 53), and twenty-six invariant sites were also observed (codons: 1–4, 6, 7, 13, 16, 21, 23–25, 30, 32, 36, 44, 45, 50, 51, 54, 57–59, 61, 62, and 64). Moreover, the Bayesian (FUBAR) and the maximal likelihood (MEME) approaches revealed that the sequences analyzed for both the family Laridae and the order Charadriiformes displayed numerous sites under diversifying and purifying selection. Although both methods pointed out a few different sites for Laridae and Charadriiformes, they usually matched each other and human PBSs.

For Laridae, both FUBAR and MEME allowed us to identify thirteen codons under positive selection, but at different positions. In both approaches, nine out of sixteen potential sites matched diversified codons of non-passerines while eight out of eighteen potential sites matched human PBSs [33,46]. In contrast, for Charadriiformes, MEME (*n* = 17) showed more sites under diversifying selection than the FUBAR approach (*n* = 14). Thirteen (MEME) and eleven (FUBAR) sites out of sixteen matched with the possible codons of non-passerines, while eleven (MEME) and seven (FUBAR) sites out of eleven matched with human PBSs [33,46]. Furthermore, the number of codons under purifying selection detected by FUBAR were 12 for Laridae and 18 for Charadriiformes; only 1 inferred site for Laridae (codon 47) matched human PBS and encoded mostly proline. The sites under significant inferred selection matched both the most conserved and polymorphic sites displayed in the alignment of MHCIIB sequences presented in other studies [33,45,46,47]. This indicates that they are probably involved in pathogen recognition. Therefore, the sites evaluated in this study may be useful to test for positive or negative historical signatures of selection in both PBR and non-PBR codons.

To detect selection, several MHCIIB studies in birds are based on the PBSs inferred for humans; however, those positions may not apply universally, and the selection is not necessarily limited to such codons [35,63]. Therefore, twenty sites (codons 8–10, 12, 16, 17, 19, 20, 26, 39, 42, 43, 48, 49, 52, 53, 56, 60, 63, and 66; Figure 5) under diversifying selection inferred by FUBAR and MEME in this study were used as PBSs in our study system and other Charadriiformes species to test the historical signals of selection imposed by pathogens. In general, for Charadriiformes, an excess of non-synonymous (*d*N) over synonymous (*d*S) changes with substantial differences (*d*N-*d*S) was detected, ranging from 0.02 ± 0.05 to 0.15 ± 0.05 in all the fragments analyzed for each species (Table 2). Two species showed less non-synonymous than synonymous changes: the Marbled Murrelet (*B. marmoratus*) and the Snowy Plover (*C. nivosus*), with differences of −0.01 ± 0.04 and −0.03 ± 0.03, respectively (*d*N-*d*S column, Table 2). Likewise, the ratios of non-synonymous over synonymous substitutions (*ω* = *d*N/*d*S) were higher at PBSs than at non-PBSs, usually rejecting the null hypothesis of strict neutrality. These results revealed significant diversifying selection (H_a_: *d*N > *d*S), acting on PBSs of twelve out of the fourteen species tested; moreover, significant purifying selection (H_a_: *d*N < *d*S) occurred in the non-PBSs of four species (*R. tridactyla*, *S. hirundo*, *B. marmoratus*, and *C. nivosus*) (Table 2). Overall, based on our inferred PBSs from the peptide-binding region (PBR), Heermann’s Gull showed lower levels of *d*N/*d*S ratio (ω = 4.5) in comparison to other species of Laridae (i.e., *R. tridactyla* ω = 12.00; *P. eburnea* ω = 6.20; and *S. hirundo*, ω = 9.75), but seemed to be low–intermediate regarding all the other Charadriiformes included in our analyses (ω = 1.66 to 12).

Using RELAX, we found strong evidence of relaxed selection in the MHCIIB alleles of Heermann’s Gull compared with other Charadriiformes (Table 3). The mechanisms by which selection can be relaxed range from removing an existing selective constraint to reducing the effective population size [73]. The estimated selection intensity (*k*) was lower when comparing Heermann’s Gull as the test set with the reference sets from two larids, the Black-legged Kittiwake (*R. tridactyla*), and the Common Tern (*S. hirundo*). The three species are listed under the categories of near threatened, vulnerable, and least concern by the IUCN red list, respectively [92]. Conversely, the selection intensity (*k*) was higher when it was compared with an endangered alcid, the Marbled Murrelet (*B. marmoratus*). In general, significant relaxed selection occurred more frequently than intensified selection when Heermann’s Gull was compared against species possessing relatively more stable populations (i.e., *R. tridactyla* and *S. hirundo*; see Table 3) [92]. Consistently, the strength of selection given by the *d*N/*d*S ratio (ω) was lower for Heermann’s Gull than for the other Larids analyzed (see *R. tridactyla*, *S. hirundo*, and *P. eburnea* in Table 2). Therefore, it is suggested that the variability of Heermann’s Gull MHCIIB may have been affected by population changes due to the direct and indirect human disturbance documented on Isla Rasa [48].

### 3.5. Demographic Expansion in Heermann’s Gull from Isla Rasa

Nucleotide and haplotype diversities were higher for MHCIIB than for cytochrome *b* (Table 4). Tajima’s *D* and Fu’s *F*_s_ statistics were lower than zero with negative values, consistent with an excess of low-frequency variants, purifying selection, or population expansion after a bottleneck event. However, only the cytochrome *b* had significant signals of purifying selection for Tajima’s *D,* while in both genes the Fu’s *F*_s_ values were significant and showed stronger signs of demographic expansion for MHCIIB than for cytochrome *b*. In contrast, the *R*_2_ statistic showed lower and significant values for cytochrome *b* but not significant values for MHC. This revealed large population growth, but only for cytochrome *b*. The Fu’s *F*_s_ is more sensitive to recent population expansion than the Tajima’s *D*, but higher Tajima’s *D* values may be a weak signal of balancing selection for MHCIIB. In general, the diversity values and the results of the neutrality tests for both genes were consistent with demographic growth [49].

The mismatch distributions under a demographic expansion model are shown in Figure 6 for both MHCIIB and the cytochrome *b* haplotypes (demographic parameters τ, *θ*_0_, and *θ*_1_ estimated for both markers are shown in Appendix A). For MHCIIB, the shape of the observed pairwise differences did not fit unimodal mismatch distribution. Demographic expansion was inconclusive, as significant goodness of fit was demonstrated for the sum of square deviations (SSD), but not for the raggedness index (*rg*). Conversely, for cytochrome *b,* the pairwise frequencies observed displayed a unimodal mismatch distribution, consistent with demographic expansion, and the goodness of fit was not significant for either *rg* or the SSD tests. On the other hand, the values for the estimated parameters were consistent with a recent expansion for MHCIIB (*t* = 28 to 31 years before the samples were collected, i.e., 2011; CI: 9–172 yr BP) than for cytochrome *b* (*t* = 48,403 to 53,781 yr BP; CI: 35,924–100,211 yr BP). The shape of distribution is sensitive to the age of the expansion, with older expansion events leading to a unimodal peak, and relatively recent expansions showing a bimodal distribution [79,80,81]. Therefore, the multimodal distribution observed for the MHCIIB samples can be attributed to a recent demographic expansion, while the unimodal peak for the cytochrome *b* could be a signal of older expansion.

### 3.6. Trans-Species Polymorphism in Charadriiformes

For codons under positive selection, the General Time Reversible (GTR) + G + I model was selected as the best model, while for codons under neutral selection the GTR + G model was selected as the best model by an SMS tool [84]. Phylogenetic reconstruction revealed that intermingled polymorphism occurred among families either in positive or neutral codons (Figure 7). However, internal branches were better supported for neutral than for positive codon trees according to the Akaike Likehood Ratio Test (aLRTs values > 0.90) [86]. Therefore, the observed MHCIIB variation was more consistent for trans-species polymorphism than for convergence. The phylogenetic tree based on positive codons had lower branch support and showed the mixed clustering of alleles according to species and family branches. This may indicate that such groups have similar selective pressures. Conversely, in the tree constructed with neutral codons, most alleles were clustered in well-supported clades, largely reflecting the taxonomic relationships at the family level (especially for Laridae and Alcidae). However, members of Scolopacide were clustered as sharing lineages with the Black-tailed Godwit (*L. limosa*) and the Ruff (*P. pugnax*). In contrast, the Snowy Plover (*C. nivosus*) was the only species analyzed for Charadriidae and clustered at a basal level in the phylogeny with a unique lineage. It is difficult to determine which alleles belong to specific loci; however, the separation of at least two lineages in many species suggests an ancient duplication event with neofunctionalization for MHCIIB in Charadriiformes (with the only exception probably occurring in the Charadriidae family).

Consistent with the trans-species polymorphism revealed in the phylogenetic tree reconstruction (Figure 7), the phylogenetic network for Charadriiformes considering synonymous and non-synonymous codons show groups with alleles from more than one family (Figure 8). Mutation events are evident. For example, the Alcidae and Laridae families have distant allele lineages, with no overlapping. Conversely, the Scolopacidae family presents some mixed clusters. Moreover, the longer lines for the Charadriidae and Scolopacidae species represent deep mutational events, which may be a signature of an ancient lineage in Charadriiformes. For example, the Snowy Plover (*Charadrius nivosus*) has only one locus [42], which could be an ancestral trait in the Charadriidae family.

The neighbor network reconstructed for the family Laridae included more alleles and showed clusters according to species; however, similar alleles were also shared between species (Figure 9). For instance, clusters consistent with species can be seen for the Common Tern (*S. hirundo*), the Black-tailed Kittiwake (*R. tridactyla*), and Heermann’s Gull (*L. heermanni*); however, the Ivory Gull (*P. eburnea*) shares similar alleles with the Black-tailed Kittiwake. In some cases, identical amino acid fragments are shared between species, i.e., between the Common Tern and Heermann’s Gull. Furthermore, it is noteworthy that more than two loci may be occurring in Laridae (this may particularly be the case of the Black-tailed Kittiwake), following the structure of the clusters between their alleles. Overall, the Heermann’s Gull alleles recovered in this study revealed signals of trans-species polymorphism when compared with other larids.

## 4. Discussion

Two clusters of MHCIIB haplotypes were found in Heermann’s Gull and considered as putative loci: DAB and DBB (Figure 1 and Appendix A). The MHCIIB is prone to duplication with neofunctionalization in avian species and can vary from low to extremely high copy numbers [33,34,35]. Based on the literature, passerines exhibit a high level of duplications [30], while non-passerines display lower duplication levels and, therefore, fewer MHC loci [33]. For instance, an extensive MHCIIB polymorphism has been reported in a passerine, the Common Yellowthroat (*Geothlypis trichas*), with a range at least 20 loci [93]. In contrast, only three different MHCIIB sequences were reported in the Galápagos hawk (*Buteo galapagoensis*), which only contained two loci [94]. Although it is not clear whether the compact organization of MHC genes with few genes in birds is ancestral, there is evidence of two ancient avian MHCIIB lineages evolving over 100 million years ago, before the radiation of all the extant birds, by a duplication event [30]. However, a significant proportion of taxonomic orders exhibit only one of the MHCIIB lineages, and current similarity between both lineages is more likely than deep phylogenetic history under diversifying selection [14,30,75]. Consistently, our results revealed at least two loci in all of the Charadriiformes species analyzed, except for the Snowy Plover (*Charadrius nivosus*) with only one locus [42]. Such differences were expected because the gene copy number can vary even between individuals of the same species [3], i.e., in the Great Snipe (*Gallinago media*) [40]. Likewise, the exhaustive criteria of the cloning conditions and the well-conserved primers used in this study [51,52,53,57,58], coupled with the proportion of individuals showing different numbers of haplotypes, point towards individual variation in the number of MHCIIB loci in Heermann’s Gull.

Notwithstanding the evidence in this study, the two loci are a minimum estimate for Heermann’s Gull; it is possible that the primers used did not amplify all of the exon 2 sequences or the genes present. To our knowledge, the MHC structure has not yet been studied in any Charadriiformes bird genomes to know how many MHCIIB loci are present. Regardless, further investigation of the MHC class II architecture of Charadriiformes is needed to elucidate their variation and confirm each locus gene expression [37]. Additionally, some haplotypes could be found only in a few individuals, possibly due to a recent duplication, but the different phylogenetic approaches employed in our study suggest that both loci are monophyletic. A similar case with two monophyletic duplicated loci has been described in Leach’s storm petrel (*Hydrobates leocorhous*) of the order Procellariiformes [90]. Thus, both Hermann’s Gull loci seem to have originated from an old duplication event followed by the accumulation of point mutations and intralocus gene conversion.

In total, our sample size of 68 Heermann’s Gull individuals showed 23 alleles at the locus DAB and 35 alleles at the locus DBB. Altogether, the differences in genetic diversity, evolutionary distances, and modeled structures by homology suggest that each locus can recognize pathogens with different affinities (Table 1, Figure 2, Figure 3 and Figure 4). The MHC class II polymorphism is correlated with the capability to recognize peptides derived from parasites through the PBR involved in presenting antigens and triggering the adaptive immune response [18,36]. Specific MHC alleles may confer a better advantage than others in confronting pathogens through balancing selection in the form of heterozygote advantage, frequency-dependent selection, or selection varying in a determined space–time [11,18]. In this way, regarding the higher polymorphism (Table 1) of DBB alleles, the shorter evolutionary distances (Table 1), the different tensions and/or amino acids (Figure 3), and the highest number of sites matching with human PBRs in comparison to DAB alleles (Figure 2 and Figure 3), it is suggested that the DBB locus could be a younger, advantageous lineage. In contrast, the DAB locus can be retained by trans-species polymorphism, as shown by intermingled alleles with other larids (Figure 7, Figure 8 and Figure 9), and its higher evolutionary distances in both PBSs and non-PBSs in comparison with the DBB cluster (Table 1). In addition, differences in the expression of MHC genes are not unusual and can vary at different levels from populations with broad geographic distributions or even within the same individual in a wide range of tissues [14,15,16]. Therefore, it would not be strange to expect that the loci of our study model are expressed at different levels and that they differ in their effectiveness against different pathogens. Our results bring about some evidence that the latter may be true. However, the sampled gulls came mainly from Isla Rasa and, although it hosts most of the studied species’ breeding population, this is not the only area of species distribution. Future MHC studies should explore more individuals from different nesting areas and from different tissues, and evaluate the possible advantages, for example, in mate choice.

Separately, each locus in Heermann’s Gull showed different substitution rates and sites under negative selection through the FEL approach, which could signal different evolutionary histories. Expectedly, because FUBAR is less restrictive than FEL [70,71], the total variation analyzed for both loci using the FUBAR approach showed significant sites under positive selection at PBSs. Positive selection is inferred as an excess of non-synonymous over synonymous substitutions under the assumption that it will occur in response to the recognition of various pathogens [18,26]. Nonetheless, the selection was not restricted to PBR codons and not all the amino acids were expected to be under positive selection. Consistently, comparing each species, we found more evidence of diversifying selection at putative PBR sites while purifying selection was often observed in non-PBR sites, even when using a conservative approach such as FEL (Figure 5). The same was revealed by our analysis of selection (Table 2); although no signs of negative selection were found in Heermann’s Gull, the negative selection was present in some other Charadriiformes (Table 2). This suggest that functional roles in putative PBR and non-PBR sites have remained very similar among species of Charadriiformes, as has also been proposed to occur among mammal species [64]. This implies that the same sites widely used as PBSs for humans can be useful for a wide spectrum of species; however, ideally, specific PBSs should be used [46,47]. This possibility does not seem too far-fetched, since the first studies with X-ray crystallography have been carried out to understand the structures of MHCIIIB in chickens [45]. Therefore, future studies should consider these structures, or others phylogenetically more related to the group under study, to obtain more reliable estimates of pathogen-mediated selection based on molecular interactions.

MHCIIB variability in Heermann’s Gull was lower and had a more relaxed selection than in similar species with more stable populations, i.e., the Black-legged Kittiwake (*R. tridactyla*) and Common Tern (*S. hirundo*) (Table 2 and Table 3). In addition to pathogen-mediated selection, micro evolutionary forces, such as genetic drift, can also change the amino acid composition of the MHC in populations [6,20]. Genetic diversity at the MHC loci may reflect the selection pressure to which a population has been exposed in its recent history [36]. Populations with higher MHC genetic diversity may persist longer through evolution [6,20,24,68,95]. Although the trans-species polymorphism pattern observed in Charadriiformes is not expected to last long within the populations, such polymorphism may be maintained under balancing selection during prolonged periods. Therefore, even after a strong selective process, and assuming there is no negative selection due to other causes, resistant polymorphisms may persist even after several generations [30,33,35,78]. Usually, MHC genes are expected to show high levels of polymorphism. Nonetheless, reduced MHC variability could be expected in small or bottlenecked populations, assuming that balancing selection may overwhelm genetic drift during recent demographic bottlenecks [9,94].

In general, the MHCIIB polymorphism found in Heermann’s Gull was concordant with signals of balancing selection and recent demographic growth (Table 4). Higher diversity and positive values for Tajima’s D were expected for MHCIIB rather than for cytochrome *b* haplotypes, considering the differences in selection and inheritance for both genes. However, the contrasting departures from neutrality for historical and recent signatures of selection, detected at cytochrome *b* and the MHCIIB, respectively, could be attributed to a strong decline between 1859 and 1975, followed by a fast demographic growth observed between 1964 and 2011 in accordance with the IUCN [48,92]. Strikingly, our estimated time for a possible recent expansion based on the MHCIIB variability of Heermann’s Gull (between 1980 and 1983; CI 95%: 1839–2002) agrees with the conservation measures undertaken on Isla Rasa since 1964, the year in which it was decreed a Nature Reserve and Bird Refuge Area [48,96]. Therefore, the MHCIIB diversity observed in Heermann’s Gull may be the result of that rapid expansion from a limited number of founding lineages.

## 5. Conclusions

This study showed lower levels of diversifying selection and more relaxed evolution in the MHCIIB of Heermann’s Gull than in other threatened species of Charadriiformes seabirds. In theory, the MHCIIB variability observed in Heermann’s Gull should have decreased because of the severe population decline that has occurred due to diverse types of anthropogenic activities in the last couple of centuries [48]. Nonetheless, such MHCIIB variability showing diversifying selection, despite only a few mutational changes, has accumulated in some individuals (i.e., unique alleles) and can be explained by a subsequent sudden demographic growth when Isla Rasa was established as a natural protected area. This, and other alternative explanations, may be debated or even refuted in the future. However, more evidence needs to be accumulated to shed light on the underlying evolutionary processes. At the present time, the MHCIIB variation observed in Heermann’s Gulls could be evidence of both historical and recent selection at two putative inferred loci. Only a few other studies have found recent signatures of selection at MHC loci in wild seabird populations despite bottlenecks (i.e., [9,94]). Thus, our results suggest the importance of the conservation measures adopted on Isla Rasa, and may be considered as an example to support conservation actions at other sites with important seabird colonies.

The evidence for trans-species polymorphism and balancing selection among Charadriiformes found in this study seems to be consistent with other studies, indicating the evolution of two ancient avian MHCIIB lineages, implying that duplication processes with neofunctionalization are vital to preserve adequate levels of variation in the avian MHC [30,31,32,33,34,35]. In addition, a set of PBSs has been proposed, which could be useful as a reference for other studies involving species closely related to the Heermann’s Gull and other Charadriiformes.

Finally, we want to emphasize the need to study MHC diversity in wild species, as research historically has focused on model species (i.e., chickens, mice, and humans), as well as stress the importance of following up on these studies for the management and conservation of species inhabiting the Gulf of California, in view of the extreme mortality and reproductive failures recently reported in seabirds in this region, as well as in other seabird species from the NE Pacific Ocean due, among other factors, to climate change and changes in oceanographic regimes [97,98,99,100,101].

## Figures and Tables

**Figure 1 genes-13-00917-f001:**
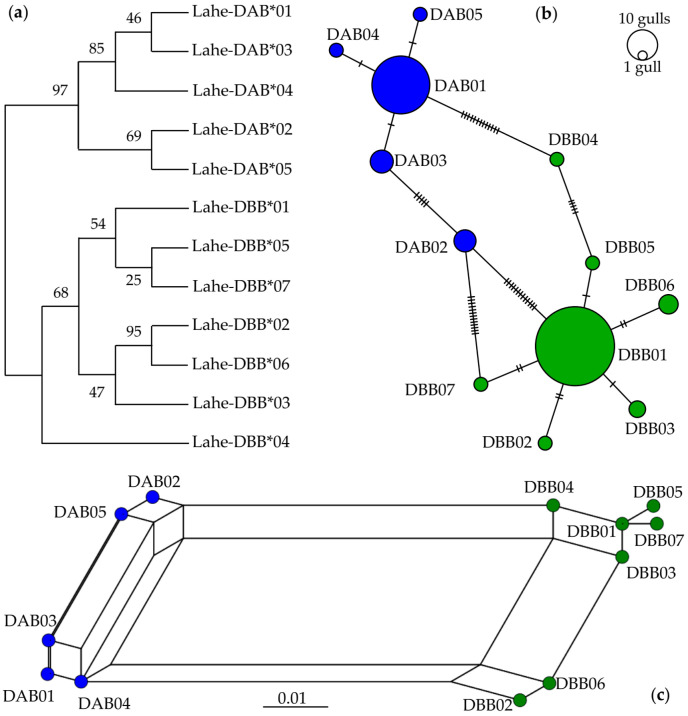
Relationships among MHCIIB alleles suggest at least two loci in Heermann’s Gull. Five and seven putative true alleles were grouped in the DAB (in blue) and DBB (green) clusters, respectively, by three approaches: (**a**) maximum likelihood tree showing the percentage of trees in which associated alleles were clustered together after 1000 bootstrap pseudoreplicates. (**b**) Minimum spanning network connecting each allele with circle sizes proportional to the number of individuals in which they were found; parallel lines show differences in base pairs. (**c**) Neighbor-net phylogenetic network exhibiting relationships between alleles with lines representing reticular (mutation) events and circles depicting each allele.

**Figure 2 genes-13-00917-f002:**
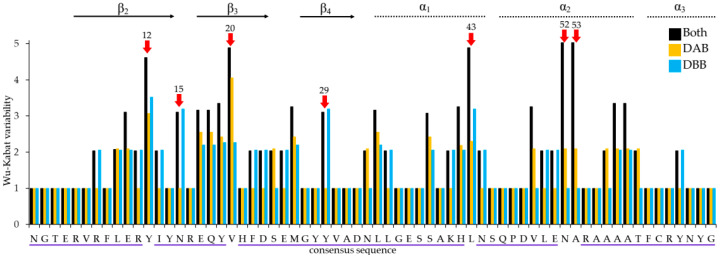
Wu–Kabat plot for the DAB and DBB clusters of Heermann’s Gull MHCIIB alleles. Red arrows indicate the most polymorphic sites, while fragments of six or more consecutive residues are underlined in purple. Black arrows and dotted lines indicate amino acid identities with inferred chicken β strands and α helices, respectively [45].

**Figure 3 genes-13-00917-f003:**
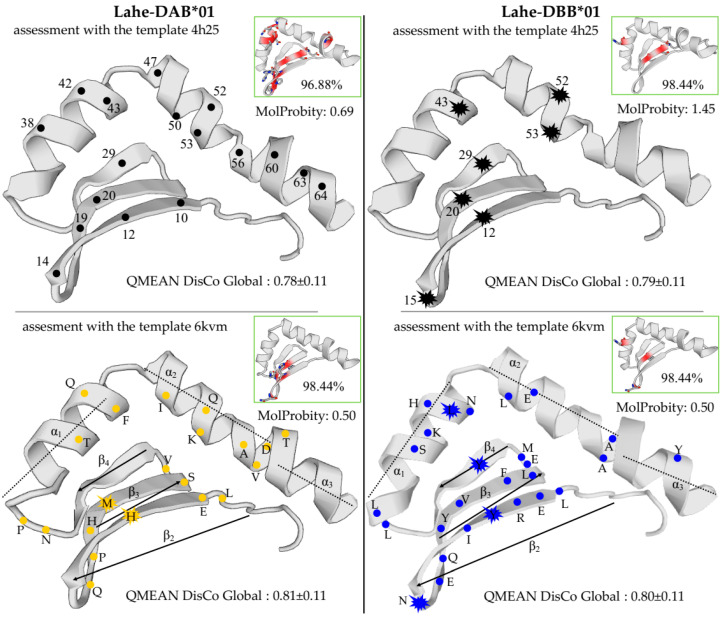
Modeling by homology for the two most common MHCIIB alleles in Heermann’s Gull (Lahe-DAB*01 and Lahe-DBB*01). The assessment lowest MolProbity scores and the highest QMEAN DisCo Global scores showed that the chicken template (6 kvm) had a better model quality than the human template (4h25). The Ramachandran-favored (%) and bad angles (in red) are shown within the green boxes. Black dots indicate the PBSs inferred from humans [46,47]. Black explosions indicate all the highly polymorphic sites inferred through Wu–Kabat variability (W). Both yellow and blue dots indicate variable sites and both yellow and blue dot explosions indicate the more polymorphic sites assessed for the DAB and DBB clusters by W variability.

**Figure 4 genes-13-00917-f004:**
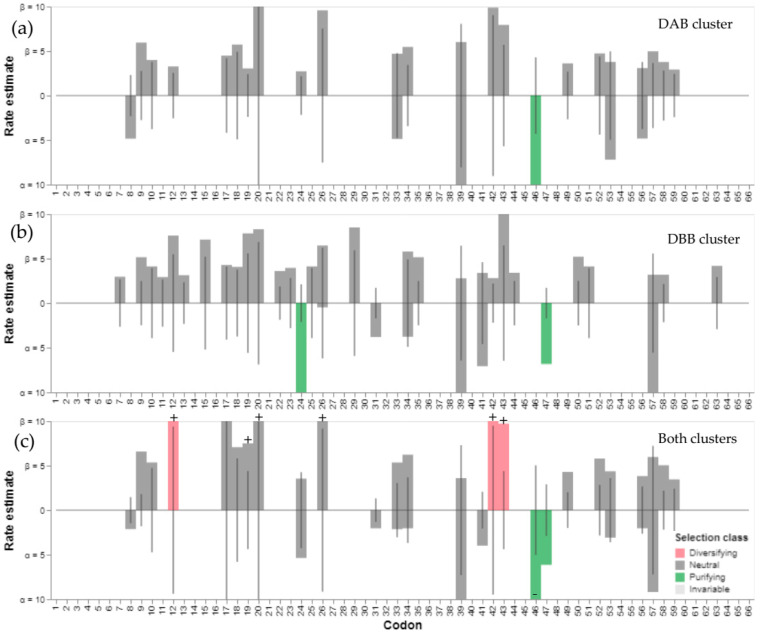
Synonymous (α) and non-synonymous (β) substitution rates assessed by the FEL approach for Heermann’s Gull MHCIIB alleles in two clusters (or putative loci): (**a**) DAB and (**b**) DBB. Estimates at each site are shown as bars and lines, indicating the estimates under the null model (α = β). The DAB cluster showed more invariant sites than the DBB cluster and only one (DAB) or two (DBB) sites displayed significant purifying selection (green bars) at each cluster. (**c**) For both clusters the maximal likelihood FEL approach revealed diversifying selection at three sites (pink bars), purifying selection at two sites (green bars), and non-significant neutral selection (gray bars) at the other twenty-one variable sites. In comparison, the Bayesian FUBAR approach indicated consistent signatures of positive selection (+) at three out of five sites and one out of two sites under negative selection (−) inferred by FEL.

**Figure 5 genes-13-00917-f005:**
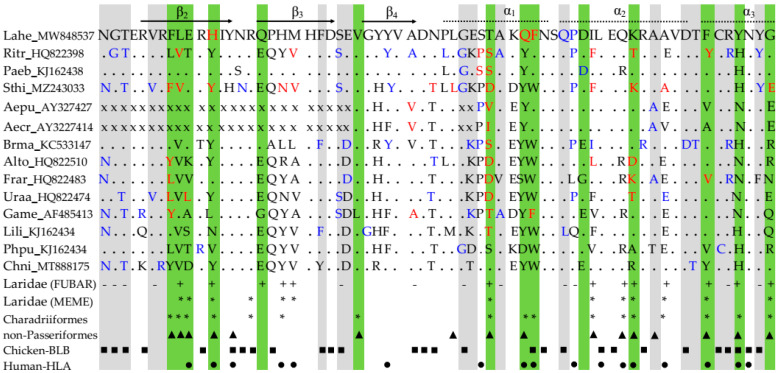
Alignment of amino acids from Heermann’s Gull MHCIIB and other Charadriiformes. Dots indicate the same residues as in Lahe_MW848537. ‘x’ represents missing amino acids. FEL assessed the sites under diversifying (in red) and purifying (in blue) selection for each species. FUBAR detected the sites under positive and negative selection separately for Charadriiformes (shaded in green and gray, respectively) and Laridae (indicated with + or − respectively). Amino acids under diversifying selection for Charadriiformes and Laridae identified by MEME (*p* = 0.05) are marked with *. The triangles show the sites under positive selection inferred for non-passerines [33], while the dots show the peptide-binding residues in humans [46,47]. The squares show conserved sites in the alignment of α and β domains (highlighted by upper arrows and dotted lines) in representative mammals and the BLB2 chain from chicken [45]. Lahe—*Larus heermanni*; Ritr—*Rissa tridactyla*; Paeb—*Pagophila eburnea*; Sthi—*Sterna hirundo*; Aepu—*Aethia pusilla*; Aecr—*Aethia cristatella*; Brma—*Brachyramphus marmoratus*; Alto—*Alca torda*; Frar—*Fratercula arctica*; Uraa—*Uria aalge*; Game—*Gallinago media*; Lili—*Limosa limosa*; Phpu—*Philomachus pugnax*; Chni—*Charadrius nivosus*. The accession number for each species shown is after the underscore.

**Figure 6 genes-13-00917-f006:**
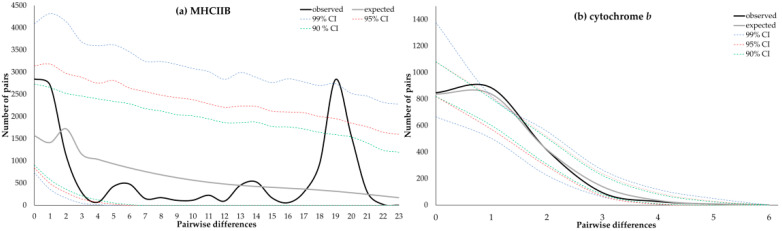
Mismatch distributions show the observed and expected frequencies of pairwise nucleotide differences from MHCIIB and cytochrome *b* haplotypes for 68 Heermann’s Gull individuals. Confidence intervals were 90% (green), 95% (red), and 99% (blue). (**a**) Multimodal distribution drawn for MHCIIB is consistent with the demographic equilibrium rather than expansion. (**b**) Conversely, the unimodal distribution depicted for cytochrome *b* is typical in populations with a recent demographic expansion.

**Figure 7 genes-13-00917-f007:**
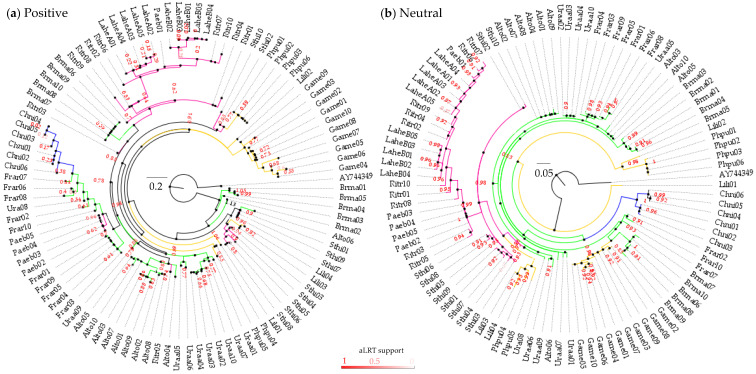
Maximum likelihood phylogenetic trees for MHCIIB alleles in Charadriiformes were inferred using positive (**a**) and neutral (**b**) codons, separately. Branch colors correspond to the family (magenta—Laridae; green—Alcidae; yellow—Scolopacidae; blue—Charadriidae). Labels in each branch show the sequences belonging to the species analyzed (i.e., Lahe = *Larus heermanni*; see methods).

**Figure 8 genes-13-00917-f008:**
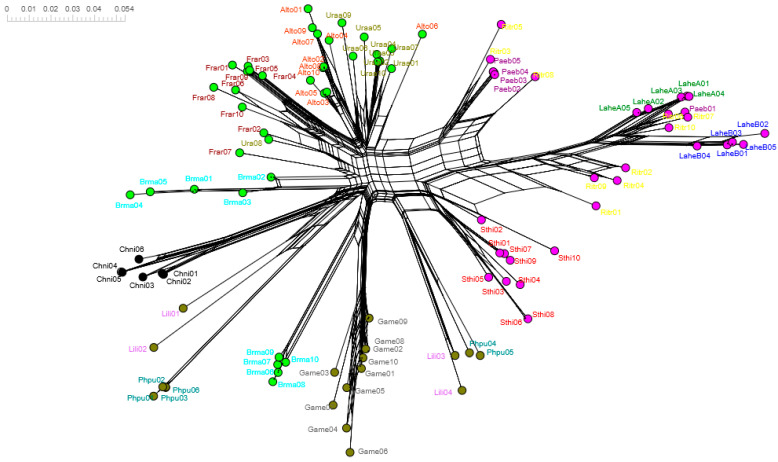
Neighbor-net phylogenetic network for MHCIIB alleles in Charadriiformes. Families Laridae (magenta dots), Alcidae (green dots), Scolopacidae (olive dots), and Charadriidae (black dots) are shown. Labels in each case show the sequences belonging to the species analyzed (i.e., Lahe—*Larus heermanni*; see methods).

**Figure 9 genes-13-00917-f009:**
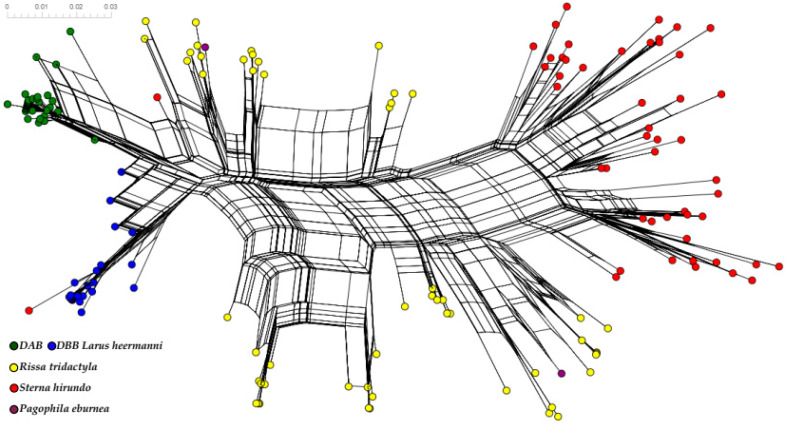
Neighbor-net phylogenetic networks for MHCIIB exon 2 alleles in Laridae species.

**Table 1 genes-13-00917-t001:** Genetic diversity of the 66 codons from the MHCIIB haplotypes of Heermann’s Gull. Codons of the peptide-binding regions (PBRs) as well as the non-peptide-binding-regions (non-PBRs) were inferred based on the human antigen-binding groove [46,47]. Both numbers of haplotypes (*n*) and codons examined are shown in parentheses.

Group of Sequences	Codons	*S*	*K*	π	*d* _nt_	*d* _aa_
*Larus heermanni* (*n* = 57)	All (66)	55	10.422	0.053	0.05 ± 0.01	0.12 ± 0.02
	PBR (17)	22	5.243	0.103	0.10 ± 0.03	0.22 ± 0.06
	Non-PBR (49)	33	5.179	0.035	0.04 ± 0.01	0.09 ± 0.02
DAB cluster (*n* = 23)	All (66)	28	4.980	0.025	0.03 ± 0.01	0.06 ± 0.01
	PBR (17)	12	2.142	0.042	0.04 ± 0.01	0.11 ± 0.04
	Non-PBR (49)	16	2.838	0.019	0.02 ± 0.01	0.05 ± 0.01
DBB cluster (*n* = 34)	All (66)	39	3.242	0.017	0.02 ± 0.00	0.04 ± 0.01
	PBR (17)	13	1.230	0.024	0.02 ± 0.01	0.07 ± 0.02
	Non-PBR (49)	26	2.012	0.014	0.01 ± 0.00	0.03 ± 0.01

*S*—segregating sites; *K*—average nucleotide differences; π—average nucleotide diversity. Evolutionary distances of nucleotide and amino acid sequences are represented by *d*_nt_ and *d*_aa_, respectively. Values after ± are standard deviations.

**Table 2 genes-13-00917-t002:** Analysis of the selected codons from the peptide-binding regions (PBRs) and the non-peptide-binding regions (non-PBRs) inferred for Heermann’s Gull MHCIIB variants and for homologous sequences (*n*) from other Charadriiformes species available in GenBank. The rates (ω = *d*N/*d*S) of the average non-synonymous (*dN*) and synonymous (*dS*) substitutions and their differences (*d*N-*d*S) are shown. Also shown are the Z-test statistics and their significance (in parentheses) to reject the null hypothesis of neutrality for the alternative hypotheses of non-neutrality (H_a_: *dN* ≠ *dS*), as well as diversifying (H_a_: *dN* > *dS*) and purifying (H_a_: *dN* < *dS*) selection.

Species	Codons	*dN*	*dS*	*dN-dS*	ω	*dN* ≠ *dS*	*dN > dS*	*dN < dS*
*Larus heermanni* (*n* = 57)	All (66)	0.07 ± 0.02	0.02 ± 0.01	0.05 ± 0.02	3.50	**2.37 (0.019)**	**2.42 (0.009)**	−2.44 (1)
	PBR (20)	0.19 ± 0.05	0.04 ± 0.03	0.15 ± 0.06	4.75	**2.71 (0.008)**	**2.61 (0.005)**	−2.82 (1)
	Non PBR	0.03 ± 0.01	0.02 ± 0.01	0.01 ± 0.02	1.50	0.56 (0.574)	0.51 (0.306)	−0.542 (1)
*Rissa tridactyla* (*n* = 53)	All	0.12 ± 0.02	0.07 ± 0.02	0.05 ± 0.03	1.71	1.62 (0.109)	**1.73 (0.043)**	−1.59 (1)
	PBR	0.36 ± 0.06	0.03 ± 0.02	0.33 ± 0.06	12.00	**5.67 (***)**	**5.39 (***)**	−5.63 (1)
	non PBR	0.04 ± 0.01	0.09 ± 0.03	−0.05 ± 0.03	0.44	−1.67 (0.098)	− 1.65 (1)	**1.71 (0.045)**
*Pagophila eburnean* (*n* = 2)	All	0.20 ± 0.04	0.10 ± 0.04	0.10 ± 0.07	2.00	1.52 (0.132)	1.42 (0.079)	−1.45 (1)
	PBR	0.62 ± 0.14	0.10 ± 0.08	0.53 ± 0.18	6.20	**2.94 (0.004)**	**2.93 (0.002)**	−2.94 (1)
	Non PBR	0.06 ± 0.03	0.10 ± 0.05	−0.04 ± 0.06	0.60	−0.65 (0.517)	−0.61 (1)	0.63 (0.265)
*Sterna hirundo* (*n* = 50)	All	0.11 ± 0.03	0.05 ± 0.01	0.06 ± 0.03	2.20	**2.44 (0.016)**	**2.36 (0.010)**	−2.44 (1)
	PBR	0.39 ± 0.09	0.04 ± 0.02	0.35 ± 0.08	9.75	**3.92 (***)**	**4.23 (***)**	−4.20 (1)
	non PBR	0.02 ± 0.01	0.05 ± 0.01	−0.03 ± 0.02	0.40	**−2.06 (0.041)**	−2.06 (1)	**2.07 (0.020)**
*Aethia pusilla* (*n* = 2)	All	0.15 ± 0.05	0.00 ± 0.00	0.15 ± 0.05	-	**2.62 (0.010)**	**2.74 (0.004)**	−2.65 (1)
	PBR	0.32 ± 0.19	0.00 ± 0.00	0.32 ± 0.18	-	1.78 (0.077)	**1.73 (0.043)**	−1.68 (1)
	Non PBR	0.08 ± 0.04	0.00 ± 0.00	0.08 ± 0.03	-	**2.13 (0.035)**	**2.11 (0.019)**	−2.01 (1)
*Aethia cristatella* (*n* = 2)	All	0.05 ± 0.03	0.02 ± 0.02	0.03 ± 0.04	2.50	0.73 (0.464)	0.78 (0.220)	−0.74 (1)
	PBR	0.12 ± 0.10	0.00 ± 0.00	0.12 ± 0.10	-	1.24 (0.216)	1.19 (0.118)	−1.20 (1)
	Non PBR	0.04 ± 0.03	0.04 ± 0.04	0.00 ± 0.05	1.00	−0.07 (0.946)	−0.07 (1)	0.07 (0.474)
*Brachyramphus marmoratus* (*n* = 20)	All	0.09 ± 0.02	0.11 ± 0.03	−0.01 ± 0.04	3.00	−0.31 (0.759)	−0.32 (1)	0.31 (0.379)
	PBR	0.29 ± 0.07	0.08 ± 0.05	0.21 ± 0.09	3.63	**2.30 (0.023)**	**2.25 (0.013)**	−2.33 (1)
	non PBR	0.03 ± 0.01	0.12 ± 0.04	−0.09 ± 0.04	0.25	**−2.10 (0.038)**	−2.12 (1)	**2.15 (0.017)**
*Alca torda* (*n* = 16)	All	0.06 ± 0.02	0.03 ± 0.01	0.04 ± 0.02	2.00	**2.52 (0.013)**	**2.46 (0.008)**	−2.49 (1)
	PBR	0.18 ± 0.05	0.04 ± 0.03	0.13 ± 0.03	4.50	**4.00 (***)**	**3.91 (***)**	−3.89 (1)
	Non PBR	0.02 ± 0.01	0.02 ± 0.01	0.00 ± 0.02	1.00	0.10 (0.918)	0.10 (0.460)	−0.10 (1.00)
*Fratercula arctica* (*n* = 19)	All	0.09 ± 0.02	0.04 ± 0.02	0.05 ± 0.02	2.25	**2.11 (0.037)**	**2.08 (0.020)**	−2.14 (1)
	PBR	0.26 ± 0.07	0.04 ± 0.03	0.22 ± 0.06	6.50	**3.37 (0.001)**	**3.57 (***)**	−3.35 (1)
	Non PBR	0.03 ± 0.01	0.04 ± 0.02	−0.01 ± 0.02	0.75	−0.61 (0.544)	−0.64 (1)	0.64 (0.263)
*Uria aalge* (*n* = 13)	All	0.10 ± 0.02	0.07 ± 0.02	0.03 ± 0.03	1.42	1.24 (0.218)	1.23 (0.112)	−1.23 (1)
	PBR	0.30 ± 0.07	0.08 ± 0.05	0.22 ± 0.08	3.75	**2.90 (0.004)**	**2.92 (0.002)**	−2.82 (1)
	Non PBR	0.03 ± 0.01	0.06 ± 0.02	−0.03 ± 0.02	0.50	−1.35 (0.179)	−1.37 (1)	1.37 (0.087)
*Gallinago media* (*n* = 35)	All	0.09 ± 0.02	0.05 ± 0.02	0.04 ± 0.03	1.80	1.13 (0.259)	1.16 (0.123)	−1.20 (1)
	PBR	0.29 ± 0.07	0.06 ± 0.04	0.23 ± 0.07	4.83	**3.26 (0.002)**	**3.35 (0.001)**	−3.27 (1)
	Non PBR	0.01 ± 0.00	0.05 ± 0.03	−0.04 ± 0.03	0.20	−1.14 (0.255)	−1.18 (1)	1.17 (0.122)
*Limosa limosa* (*n* = 4)	All	0.16 ± 0.03	0.12 ± 0.03	0.05 ± 0.03	1.33	1.30 (0.195)	1.31 (0.096)	−1.32 (1)
	PBR	0.31 ± 0.10	0.10 ± 0.06	0.21 ± 0.08	3.10	**2.59 (0.011)**	**2.75 (0.004)**	−2.75 (1)
	Non PBR	0.11 ± 0.02	0.13 ± 0.04	−0.02 ± 0.04	0.84	−0.46 (0.643)	−0.45 (1.00)	0.45 (0.325)
*Philomachus pugnax* (*n* = 4)	All	0.16 ± 0.03	0.13 ± 0.04	0.02 ± 0.05	1.23	0.52 (0.604)	0.49 (0.311)	−0.49 (1)
	PBR	0.27 ± 0.10	0.08 ± 0.08	0.19 ± 0.10	3.38	1.88 (0.063)	**1.84 (0.035)**	−1.75 (1)
	Non PBR	0.11 ± 0.03	0.16 ± 0.06	−0.04 ± 0.05	0.69	−0.78 (0.440)	−0.77 (1)	0.79 (0.217)
*Charadrius nivosus* (*n* = 4)	All	0.04 ± 0.02	0.07 ± 0.03	−0.03 ± 0.03	0.57	−1.11 (0.271)	−1.12 (1)	1.35 (0.129)
	PBR	0.10 ± 0.05	0.06 ± 0.05	0.04 ± 0.03	1.66	1.26 (0.211)	1.30 (0.099)	−1.26 (1)
	Non PBR	0.02 ± 0.01	0.08 ± 0.04	−0.06 ± 0.04	0.25	−1.54 (0.124)	−1.66 (1)	**1.71 (0.045)**

Standard errors were obtained using 1000 bootstraps. Significant results (*p* ≤ 0.05) for the *Z*-tests are shown in bold. *** *p* ≤ 0.0005.

**Table 3 genes-13-00917-t003:** Estimates of the selection intensity (*k*) recovered with the RELAX approach. Effects of intensified (*k* > 1) or relaxed (*k* < 1) selection were inferred for the MHCIIB variants of Heermann’s Gull (near threatened) from this study, comparing it separately with homologous sequences for thirteen species of Charadriiformes in GenBank as reference branches. Significant effects are shown in bold (*p* < 0.05). The International Union for Conservation of Nature (IUCN) red list category is shown for each species.

Reference Branches	*k*	Selection	*p*	IUCN
*Rissa tridactyla* (71)	0	**relaxed**	0.000	Vulnerable
*Sterna hirundo* (55)	0	**relaxed**	0.000	Least concern
*Pagophila eburnea* (5)	1.11	intensified	0.773	Near threatened
*Aethia pusilla* (2)	0.28	**relaxed**	0.016	Least concern
*Aethia cristatella* (2)	1.01	intensified	0.979	Least concern
*Brachyramphus marmoratus* (27)	1.77	intensified	0.126	Endangered
*Alca torda* (24)	0.78	relaxed	0.342	Near threatened
*Fratercula arctica* (33)	0.71	**relaxed**	0.048	Vulnerable
*Uria aalge* (14)	0.55	relaxed	0.093	Least concern
*Gallinago media* (37)	0.44	**relaxed**	0.000	Near threatened
*Limosa limosa* (4)	1.14	intensified	0.617	Near threatened
*Philomachus pugnax* (6)	0.75	relaxed	0.201	Least concern
*Charadrius niveus* (6)	0.32	**relaxed**	0.000	Near threatened

The number of sequences employed are shown in parentheses.

**Table 4 genes-13-00917-t004:** Genetic diversity, demographic parameters, and mismatch distributions with goodness of fit for MHCIIB and cytochrome *b* haplotypes, respectively, were recovered from 68 Heermann’s Gulls.

Genes	π	*Hd*	*D*	*F* _s_	*R* _2_	τ	*t*	*rg*	SSD
MHCIIB	0.0473	0.82	−0.16 ns	−21.70 *	0.08 ns	19.82 (6.44–108.83)	28 to 31 yr BP (9–172 yr BP)	0.04397 *p* = 0.1564	0.06017 *p* = 0.0396
Cyt-*b*	0.0009	0.63	−2.12 *	−11.06 *	0.04 *	1 (0.74–1.86)	48,403 to 53,781 yr BP (35,924–100,211 yr BP)	0.06358 *p* = 0.1782	0.00086 *p* = 0.6119

Average nucleotide diversity (π), haplotype diversity (*Hd*), Tajima’s *D*, Fu’s *F*_s_, and *R*_2_ statistics are shown. * *p* < 0.05; ns—nonsignificant values. The confidence intervals were calculated by 10,000 randomizations of the data to estimate the time since the putative expansion (*t*) event using the demographic parameter tau (τ = 2*ut*) (yr BP—years before present, regarding the year of sampling that was 2011). Lower (2.5%) and upper (97.25%) boundaries are shown in parentheses. Goodness of fit was tested by raggedness (*rg*) and sum of square deviations (SSD).

## Data Availability

The data presented in this study are available in Appendix A. Additionally, can be found at the GenBank (https://www.ncbi.nlm.nih.gov/ accessed on 18 may 2022), or requested from the corresponding authors.

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
