# Peer review of "Characterization, Selection, and Trans-Species Polymorphism in the MHC Class II of Heermann’s Gull (Charadriiformes)"

_genes, 2022, doi:10.3390/genes13050917_

Round 1
Reviewer 1 Report
This manuscript examined the variability of Heermann’s Gull Major histocompatibility complex class II (MHCIIB) and compared these loci with other Charadriiformes. It found that (1) fifty-nine MHCIIB haplotypes were identified in 68 Heermann’s Gulls: twelve as putative true alleles, forty-five as unique alleles, and two as pseudogenes; (2) there were at least two loci in Heermann’s Gull MHCIIB and trans-species polymorphism among Charadriiformes; (3) Those sites under diversifying selection revealed a better match with peptide-binding sites inferred in birds. Heermann’s Gull showed MHCIIB variability consistent with a population expansion. Duplication contributes to shaping MHCIIB variability in Charadriiformes, which enables to buffer selective pressures through balancing selection. Generally, this study is well designed and conducted. The manuscript is well-written, contributing to our understanding of the polymorphism, functional peptide sites, and evolution of the MHC gene in Charadriiformes.
Some specific comments:
As far as I am concerned, the manuscript could be simplified to remove those unnecessary explanations. Some basic information in methods and discussion could be integrated into the introduction as background to strengthen the importance of the study, and why it is designed based on known knowledge concerning target species or groups, as well as methods.
Abstract: Line 25: indicated at least
Introduction: It is also necessary to introduce a little bit of why this species is important (why chose this species but not others) for this study in the last paragraph of the introduction.
Methods: Lines 111-113 & Line 136 & Line 279-280: The general introduction of the studied species or the duplication of MHCIIB genes in Charadriiformes [33] can be integrated into the introduction as background information.
Line 218: Remove “Note:” and start from “All the analyzed MHCIIB sequences from other studies were downloaded and are available in GenBank”.
Author Response
We appreciate your comments and suggestions, which were very concise and helped to improve the work. We have amended some basic information, and all changes were highlighted in the main text using Word's change tracking tool. Following the specific comments, information from other sections was integrated into the last paraph of the introduction as background to point out the importance of the study.
Abstract: Line 25: the typo was changed by ‘indicated at least’
Line 218: “Note:” was removed and now the sentence starts from line 221 as “All the analyzed MHCIIB sequences from other studies were downloaded and are available in GenBank”.
Reviewer 2 Report
The manuscript is well written and clear even if it is a little overly long in some parts, which would have been shorter without loosing the information they deliver.
Materials and methods are well suited to achieve the study goal even though I think that the interpretation of mismatch distributions might be improved: for instance, do authors consider the two duplicated genes (DAB and DBB) jointly or separately when analysing pairwise mismatch distributions as well as neutrality tests? Moreover, bi-modal mismatch distributions also arise as a result of secondary contact, which however does not seem to be the case in this study.
Overall, the English language seems fine, but a final check for misspellings would be welcome. Moreover some statements need to be slightly improved to achieve better clarity. Some examples are given below:
Lines 631-632: This sentence need to be slightly rephrased to achieve better clarity
Line 655: Replace "In accord with literature" with "Based upon literature"
Line 682: "An example of this latest occurring has been described" should be rephrased as "A similar case, two monophyletic duplicated loci, has been described in..." If my statement's interpretation is right, isn't it?
Line 684: Replace "accumulated" with "the accumulation of", please
Line 695: Relace "In this way" with "This way", please
Lines 745-748: Maybe I am missing something, but the more I read this statement, the more I think the opposite of what authors stated holds true: in bottlenecked populations, balancing selection may overcome genetic drift, thus allowing for te maintenance of intermediate to high levels of variation, isn't it? Please, explain
Some figures and table captions miss the explanation of some acronyms. It could be hard for the reader move bacj and forth through the text to retrieve their meaning. Ideally, Figure and tables with their caption should be self-explanatory, as much as possible
Author Response
We are grateful for your helpful review; the comments and suggestions were pertinent. We understand and respect the reviewer point of view and agree that some parts of the document are long. To amend overly long parts, we have removed some lines, which were integrated into the last paragraph of the introduction to highlight the importance of the study. Word's change tracking tool highlights all changes in the main text.
Regarding the question of how mismatch, as well as neutrality tests, were analyzed, all the MHC variants found were considered jointly. The data of the variants of the MHCII found in each Heermann’s Gull individual are in the supplementary material (Supplementary Material 5). Furthermore, for all individuals the sequences labels show whether they belong to DAB or another DBB group. Therefore, if someone wants to test our data, they should obtain very similar results to those reported in the manuscript. In our case, we consider that all MHCII variants should be subject to the same selection pressures since they participate specifically in the recognition of pathogens, for this reason, they were analyzed together.
In addition to what you mention about bimodal distributions, these may be arisen because of secondary contact. Bimodal mismatch distributions are characteristic of species where two populations were separated by a geographic barrier (Horne et al. 2008, p. 636), also have been suggested to be consistent with divergence followed by population growth (Sousa et al. 2012, p. 527), or indicate the presence of two lineages (Jenkins et al. 2018, p. 10), both latest interpretations seem to make more sense to the results observed in our analyses. This way, the first peak would represent intra-lineage pairwise differences (Jenkins et al. 2018, p. 10), while the second peak would probably represent pairwise differences between ancient clades.
We thank you for all the minor changes. We agree that some parts of the text need revision. Therefore, have modified or added the following:
Before lines, 631-632 are now rephrased in lines 636-638 as “Moreover, the longer lines for Charadriidae and Scolopacidae species represent deep mutational events, which may be a sign of an ancient lineage in Charadriiformes”.
Before line 655, now line 662: "In accord with literature" was replaced with "Based upon literature"
Before 682, now 689: "An example of this latest occurring has been described" was rephrased as "A similar case, two monophyletic duplicated loci have been described in..."
Before lines 746-748, now lines 753-756: “Also, the intermediate to the high variability of MHC observed in populations that have gone through a bottleneck, may indicate that genetic drift may overcome balancing selection during recent demographic bottlenecks [9,94].” was changed by “Usually, MHC genes are expected to show high levels of polymorphism. Nonetheless, reduced MHC variability could be expected in small or bottlenecked populations, assuming that balancing selection may overwhelm genetic drift during recent demographic bottlenecks [9,94].”
Figures and tables captions acronyms were corrected.
References:
- Horne, J.B.; van Herwerden, L.; Choat, J.H.; Robertson, D.R. High population connectivity across the Indo-Pacific: Congruent lack of phylogeographic structure in three reef fish congeners. Mol. Phylogenet. Evol. 2008, 49, 629–638, doi:10.1016/j.ympev.2008.08.023.
- Jenkins, T.L.; Castilho, R.; Stevens, J.R. Meta-analysis of northeast Atlantic marine taxa shows contrasting phylogeographic patterns following post-LGM expansions. PeerJ 2018, 2018, doi:10.7717/peerj.5684.
- Sousa, L.C.C.; Gontijo, C.M.F.; Botelho, H.A.; Fonseca, C.G. Mitochondrial genetic variability of Didelphis albiventris (Didelphimorphia, Didelphidae) in Brazilian localities. Genet. Mol. Biol. 2012, 35, 522–529, doi:10.1590/S1415-47572012005000035.